# Knowledge Spillover through Blockchain Network in Tourism: Development and Validation of Tblock Questionnaire

Spyros Avdimiotis * and Panagiotis Moschotoglou

Department of Organization Management, Marketing and Tourism, International Hellenic University, 57400 Thessaloniki, Greece; dommt00920@ommt.ihu.gr
* Correspondence: soga@ihu.gr

**Abstract:** With the advent of disruptive technologies, blockchain is considered to be the most prominent technology that has the potential to have a significant influence on the knowledge management field, particularly knowledge sharing, knowledge transfer and knowledge spillover. This paper outlines the development and validation of the Knowledge Spillover through Tourism Blockchain Questionnaire, the TBlockQ. The purpose of this questionnaire is to acknowledge the key factors aligned with the level of knowledge spillover generated within a blockchain network. The TBlockQ was a synthesis of 29 5-point Likert scaled questions. A total of 422 correspondents participated in this study. The results of this study outline the reliability and validation of this questionnaire. The reliability statistics of all the items are high. Three factors, revealed from the factor analysis, identifying the knowledge spillover from a blockchain network in the tourism industry were: (a) networking expansion, improvement and spillover intention, (b) entrepreneurial and future prospects and (c) trust and security.

**Keywords:** knowledge spillover; blockchain technology; questionnaire; validation; factor analysis





## 1. Introduction

Knowledge has been widely recognized as a continuous, dynamic and multidimensional process, involving the overlapping acquisition and transfer actions, with significant emphasis on the development of competitive advantage and development of organizations, in general [1–5]. With the advent of disruptive technologies in the uprising digitalization era, the impact of information technology is commonly cited as a solution not only for knowledge sharing [6] but also for knowledge acquisition and knowledge spillover. A blockchain is considered a decentralized and distributed network that uses cryptographic hashes to store transactions among peers. The key components of blockchain technology are immutability, transparency, decentralization [7], enabling tamper-proof, transparent, and cryptographic transactions [8], while its decentralized nature eliminates the need for a central authority [9]. Blockchain technology offers the potential to have a significant influence on knowledge management, particularly knowledge sharing, knowledge transfer and knowledge spillover.

Knowledge transfer and spillovers are positively related to blockchain startups and various sectors (e.g., supply chain, tourism, finance, medical, etc.) [10]. The issue of the whitepapers plays a vital role, not only as a marketing tool [11] but also as a knowledge transfer tool for other startups to acquire knowledge. Another aspect of the knowledge spillovers across the blockchain industry is the open-source algorithms that startups upload on their main websites. An array of blockchain startups relies on their own blockchain structure, creating their own digital wallets, smart contracts and even their own cryptocurrencies. All of these algorithms are uploaded on their main website, in order to be accessible to other startups that want to build their network based on these same algorithms. This phenomenon not only provides intentional knowledge transfer among

other entrepreneurs but there is also an intentional spillover of knowledge among startups and future entrepreneurs. In the tourism sector, blockchain is gradually becoming the cornerstone of future development and competitiveness [12]. Blockchain is becoming the evolution of Web 2.0, expanding the interaction between peers and the ability to exchange value and perform not only safe and decentralized transactions but also exploitable and specific knowledge.

When trying to investigate the connection between knowledge spillover and blockchain networks, a questionnaire (as a research instrument) was developed to facilitate the research intention. A review of the literature (using the VOSViewer software v. 1.6.18, Leiden University, Leiden, The Netherlands) showed that there was a lack of such an instrument to measure knowledge spillover in finite networks. The items structuring the questionnaire are based on the SECI Model knowledge transfer [1] and the basic preferences of finite networks, such as a blockchain. To this end, the object of this paper was the development and validation of the Knowledge Spillover through Tourism Blockchain Questionnaire (TBlockQ). The TBlockQ was developed in order to measure the level of knowledge spillover generated within a blockchain network in the tourism industry.

Blockchain technology research in Greece follows a pattern of innovation readiness. According to the Global Innovation Index (2021), Greece was ranked 47th among 132 countries, indicating that Greece is more a follower than a creator of innovation. Blockchain technology is still in its infancy, however, paradigms of implementation are taking place, not only in research labs but also in sectors, such as wineries, supply chains and tourism.

This paper is divided into four distinctive parts. In Section 1, the authors attempt a literature overview of blockchain technology in general, and the knowledge spillover effect. Section 2 regards the methodology used to validate the questionnaire. Section 3 focuses on the statistical analysis of the research findings, while in Section 4, survey findings are discussed, pinpointing the three factors generated from the factor analysis.

## 2. Literature Review

### 2.1. Overview of the Blockchain

A blockchain is a decentralized and distributed database, which contains and stores a set of digital records and transactions, which are interconnected together in a list of blocks by using cryptography [13]. The term blockchain was introduced in 2008 by an anonymous person or a group of people known as Satoshi Nakamoto in a whitepaper entitled, "*Bitcoin: A Peer-to-Peer Electronic Cash System*" [14]. The author(s) laid out the framework for blockchain without actually using the term "Blockchain" in their whitepaper; instead describing a detailed method of how to accomplish transactions without depending on trust—stemming from the existence of a monetary entity (e.g., a bank)—and preventing double-spending [14]. Every node in the network accepts the block with the transaction, but only if this transaction is valid. The first block on the blockchain network is called the genesis block, and each block consists of a block header and the block body. Each user on the network owns a pair of private and public keys [15]. The private key is used in order to sign the transactions, while the public key is visible to everyone in the network when the transaction is considered valid.

The key components of blockchain technology enable an array of applications among multiple industries and fields, such as tourism, supply chains, etc. [7]. The most interesting aspects of blockchain technology are its immutability and decentralization [16]. Immutability provides non-repudiation of the stored data [9], and traceability. According to [7], the immutability of the blockchain transforms the "Internet of Information", in which digital data can be copied without loss of accuracy into the "Internet of Value", in which digital data can be transferred between peers without the need for intermediation and double-spending. Decentralization provides two peers on a blockchain the ability to make transactions without the authentication from an intermediate [15]. Not only does decentralization eliminate the need for intermediation, but it also reduces the cost of transactions. Transparency is another aspect of blockchain technology, allowing peers to have access

to read data. Yet, blockchain technology, or rather a combination of technologies called distributed ledger technologies, is still under development; however, programmability plays an important role as an aspect of blockchain technology because it makes every blockchain more powerful. Consensus protocols minimize the need for trust between two peers using consensus algorithms, such as proof-of-work (PoW) [9].

The smart contract is reshaping not only blockchain as a technology by making it more powerful, but industries and business processes as well [17] and was proposed by Nick Szabo in 1997 [18]. This type of digital contract is a self-enforcing and self-executing algorithm which provides an agreement between peers [19]. Smart contracts are capable of influencing business models by eliminating the need for intermediation [20,21], improving the efficiency of business processes, and reducing risks and costs [17]. Furthermore, the smart contract algorithm cannot be modified once deployed in the blockchain network [9].

Taking account of the aforementioned, it could be stated that, in the era of digital transformation, information technology plays a vital role in solutions for inter-sectoral and intra-sectoral industry knowledge sharing. Blockchain technology could facilitate knowledge transfer, offering the ability to receive useful, exploitable, transparent and secure knowledge. In alignment, [22] developed Knowledge Blockchains, in order to store knowledge in an immutable and transparent way, while [6] proposed a blockchain-based knowledge system, which provides transparency, trust and time-effectiveness in knowledge creation, transfer and sharing.

### 2.2. Knowledge Spillover

Knowledge spillover refers to the unintentional transfer of knowledge among two or more peers. At every possible interaction, either individual or organizational, there is a huge potential for knowledge exchange [5], while in the case of intentional conveying, knowledge spillover turns into knowledge transfer. The ability of a person to absorb knowledge, either intentionally (transferred) or unintentionally (spillover), depends on the person's absorptive capacity [23]. The ability of absorptive capacity, according to Tidd [24], depends on the ability of an organization to absorb knowledge, depending on the degree of experience it acquires, as well as the degree of training of its human resources.

There are three levels of knowledge spillover. First, there is the individual level, where knowledge is unintentionally transferred between people and could happen when tacit knowledge is externalized without realizing it [23]. Second, knowledge could be transferred within an organizational level, where organizations with close geographical distance could either transfer intended knowledge or spillover unintended knowledge. Organizations and industries that are geographically close to each other can benefit most from knowledge spillover. However, with the advent of technology, knowledge can be transferred across borders without restrictions. The MAR theory of spillovers, established by Marshall (1920), Arrow (1962) and Romer (1986) [25–27], relies on the intra-sectoral spillovers within industries. The knowledge gathered by one company tends to help other companies' technologies without proper compensation [28]. Inter-industry knowledge spillovers occur as a result of the diversification of knowledge as well as diversity between supplementary industries or customers and suppliers serving each other [29]. On the contrary, inter-sectoral spillovers are based on Jacobs' [30] theory of knowledge spillovers externalities. According to Braunerhjelm [31], knowledge spillover outside the core industry is expanding geographically among cities and even countries. Based on the above acceptance of the theory of knowledge spillover, the third level of knowledge spillover takes place at an international level. Geographic proximity and diversity are important aspects in the transmission of knowledge because knowledge is inherently non-competitive in nature and knowledge was developed for any specific application, maximizing the potential for innovation and growth globally [32].

### 3. Research Methodology

The primary objective of this study was to develop a questionnaire to measure the spillover of knowledge in a blockchain network. The overall structure of the TBlockQ was a synthesis of an adequate number of 5-point Likert scaled questions (N = 29), with answers varying from 1 = Absolutely Disagree to 5 = Absolutely Agree. Based on the SECI model of knowledge transfer [1] and the key components of the blockchain technology, namely immutability, the decentralized and distributed nature of it, and the trust and security of the transactions. A number of questions were also added in order to include the aspect of a blockchain network and the communication between the nodes. The distribution of the questionnaire was administrated via email. The questionnaire was distributed between March and May 2021.

A total of 422 correspondents familiar with blockchain technology, 32% were women and 68% were men, participated in this study. Correspondents' ages ranged from 18 to 59 years, whereas the majority of the subjects ranged from 30 to 44 years (SE = 0.055, SD = 0.607). The majority of the correspondents held a Master of Science degree (50.8%), while the other 49.2% have an undergraduate degree. A total of 55.7% of the correspondents work in the private sector, 6.6% in the public sector, 23.8% of the correspondents are entrepreneurs, and 13.9% of the correspondents are still university students.

The object of the survey was to develop a valid questionnaire to be used to acknowledge the key factors associated with knowledge spillover in a blockchain network.

Validation is a task requiring accurate statistical analysis to determine whether the questionnaire measures accurately the notions of knowledge spillover in a peer-to-peer ecosystem. In this aspect, Garcia et al. [33] postulate that a valid questionnaire must be simple, understandable, feasible, reliable, precise, and finally, can obtain content and construct coherence. Despite those references, the very subject of a questionnaire's validation widely varies regarding the set of statistical tools and techniques required. In this case, validation was based on a validation scheme, including (a) Cronbach's alpha test and (b) factor analysis (FA). In more detail, values distribution normality was primarily checked following this methodology; weights were also acknowledged allowing the performance of composite reliability (CR) and average variance extracted (AVE), which, according to [34], evaluates the correlation with model items.

### 4. Results and Discussion

#### 4.1. Reliability of the Questionnaire and Statistical Analysis

Cronbach's alpha and item-total correlation coefficients were used to examine the reliability of the Knowledge Spillover through Blockchain Technology Questionnaire. George and Mallery [35] proposed four reliability cut-off points: high reliability (0.90 and above), good reliability (0.70–0.90), moderate reliability (0.50–0.70), and low reliability (0.50 and below). All 28 items of this questionnaire reached a Cronbach's alpha coefficient of 0.93. In this regard, the threshold of 0.93 was exceeded, indicating that high reliability and internal cohesion of the questionnaire's items were achieved. Regarding the score of average variance extracted (AVE) and composite reliability (CR), the following (Table 1) indicates the validity of the items' load factors, their average loading and the composite reliability score. Cronbach's alpha, AVE and CR values imply the discriminant validity of the questionnaire, meaning that the items used are coherent and associated.

**Table 1.** AVE and CR measurements.

| | |
|---|---|
| Total Average Variance Extracted (AVE) | 0.555 |
| Total Composite Reliability (CR) | 0.931 |

Factor analysis is frequently used in questionnaire development, aiming to determine the existence of relationships between the items in a questionnaire, as well as any clustering effects between them. The fundamental principle of factor analysis is that items should be loaded on specific underlying factors that are highly correlated with the questionnaire

items. The Kaiser–Meyer–Olkin statistic adequacy assessment was initially performed to determine whether factor analysis should be employed to interpret data from the Knowledge Spillover through Blockchain Technology Questionnaire. The Kaiser–Meyer–Olkin measure of sampling adequacy was 0.685 (Table 2). Factor analysis was conducted to analyze the data because of the high Kaiser–Meyer–Olkin ratio.

**Table 2.** The Kaiser–Meyer–Olkin Measure of Sampling Adequacy.

| Kaiser–Meyer–Olkin Measure of Sampling Adequacy | | 0.713 |
|---|---|---|
| Bartlett's Test of Sphericity | Approx. Chi-square | 3802.408 |
| | df | 378 |
| | Sig. | 0.000 |

The factor structure of the Knowledge Spillover through Blockchain in Tourism Questionnaire was evaluated using the principal component analysis extraction technique. Three factors were generated by factor analysis explaining, respectively, 79.066% of the total variable. The first factor explained 50.3% of the total variance; the second factor explained 15.01% of the total variance and the third factor explained 13.7% of the total variance.

The factor loadings were rotated using varimax rotation with Kaiser normalization to make the data easier to comprehend (this technique spreads the variation more equally across the three components). Analyzing the three extracted factors separately, 15 items successfully loaded on the first factor, which was described as "Networking expansion, improvement and spillover intention", 9 items loaded on the second, which was labeled as "Entrepreneurial and future prospects", and 5 items loaded on the third factor, which was interpreted as "Trust and security" (see Figure 1).

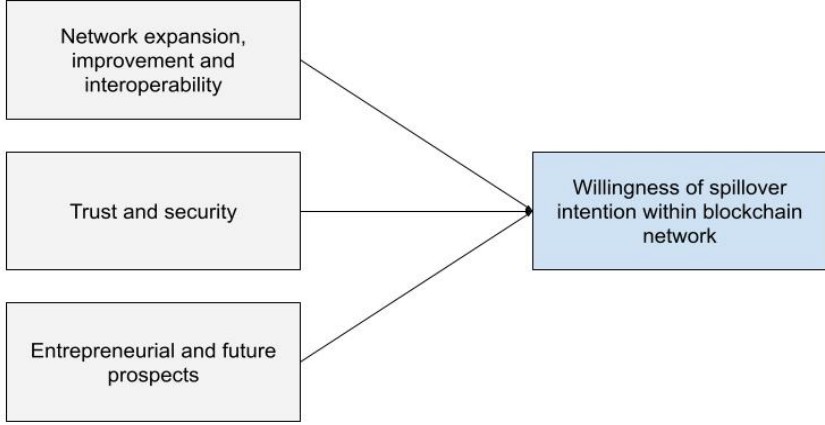

**Figure 1.** Factors explaining the willingness of spillover intention within a blockchain network.

The AVE and CR loadings were also measured per extracted factor. The analysis showed that both AVE and CR were within the appropriate values, and more significantly for the "Networking expansion, improvement and spillover intention" factor, the CR was 0.94 and the AVE was 0.50. For the factor "Entrepreneurial and future prospects", the CR was 0.92 and the AVE was 0.57, and for the factor "Trust and Security", the CR reached the value of 0.91 and the AVE was 0.68 (Table 3).

**Table 3.** Items Loadings, Average Variance Extracted and Composite Reliability.

| Factors | Items | Estimate | Square Loadings | Sum of Squared Loadings | AVE | Delta | Sum of Loadings | Sum of Loadings Squared | Sum of Delta | CR Denominator | CR |
|---|---|---|---|---|---|---|---|---|---|---|---|
| Networking Expansion and Improvement and Spillover Intention | F.1.1 | 0.916 | 0.840 | | | 0.160 | | | | | |
| | F.1.2 | 0.524 | 0.270 | | | 0.730 | | | | | |
| | F.1.3 | 0.884 | 0.780 | | | 0.220 | | | | | |
| | F.1.4 | 0.862 | 0.740 | | | 0.260 | | | | | |
| | F.1.5 | 0.823 | 0.680 | | | 0.320 | | | | | |
| | F.1.6 | 0.664 | 0.440 | | | 0.560 | | | | | |
| | F.1.7 | 0.626 | 0.390 | | | 0.610 | | | | | |
| | F.1.8 | 0.546 | 0.300 | | | 0.700 | | | | | |
| | F.1.9 | 0.742 | 0.550 | | | 0.450 | | | | | |
| | F.1.10 | 0.591 | 0.350 | | | 0.650 | | | | | |
| | F.1.11 | 0.755 | 0.570 | | | 0.430 | | | | | |
| | F.1.12 | 0.732 | 0.540 | | | 0.460 | | | | | |
| | F.1.13 | 0.518 | 0.270 | | | 0.730 | | | | | |
| | F.1.14 | 0.620 | 0.380 | | | 0.620 | | | | | |
| | F.1.15 | 0.651 | 0.420 | 7.53 | 0.500 | 0.580 | 10.45 | 109.29 | 7.47 | 116.76 | 0.940 |
| Entrepreneurial and Future Prospects | F.2.1 | 0.850 | 0.720 | | | 0.280 | | | | | |
| | F.2.2 | 0.896 | 0.800 | | | 0.200 | | | | | |
| | F.2.3 | 0.854 | 0.730 | | | 0.270 | | | | | |
| | F.2.4 | 0.686 | 0.470 | | | 0.530 | | | | | |
| | F.2.5 | 0.563 | 0.320 | | | 0.680 | | | | | |
| | F.2.6 | 0.565 | 0.320 | | | 0.680 | | | | | |
| | F.2.7 | 0.714 | 0.510 | | | 0.490 | | | | | |
| | F.2.8 | 0.813 | 0.660 | | | 0.340 | | | | | |
| | F.2.9 | 0.790 | 0.620 | 5.16 | 0.570 | 0.380 | 6.73 | 45.31 | 3.84 | 49.15 | 0.920 |
| Trust and Security | F.3.1 | 0.879 | 0.770 | | | 0.230 | | | | | |
| | F.3.2 | 0.931 | 0.870 | | | 0.130 | | | | | |
| | F.3.3 | 0.649 | 0.420 | | | 0.580 | | | | | |
| | F.3.4 | 0.805 | 0.650 | | | 0.350 | | | | | |
| | F.3.5 | 0.823 | 0.680 | 3.39 | 0.680 | 0.320 | 4.09 | 16.70 | 1.61 | 18.32 | 0.910 |

*4.2. Discussion*

In the first extracted factor, all 15 item loadings were exceeding the threshold of 0.5; the KMO score was also within the acceptable range, indicating the validity of the outcomes. However, in the factor analysis, the most important and crucial part is to identify the factors and suggest the common meanings between the items of the questionnaire. More specifically, the first factor was named "**Networking expansion, improvement and spillover intention**", and was portrayed by the following items:

1. I would like the Blockchain network to expand and I am willing to participate in its development.
2. I intend to share information with the nodes of the blockchain network.
3. A blockchain network can be a hub for participatory knowledge.
4. The interoperability of blockchains will contribute to the better knowledge spillover between the nodes.
5. The knowledge I gained through blockchain will help me develop future applications.
6. I discuss with my colleagues/partners ideas I have drawn from successful apps based on blockchain technology.
7. In a blockchain network all nodes have adequate access.
8. Within the blockchain network I have access to white papers of institutions and companies.
9. In a blockchain network I know where to look to find the right piece of information
10. White papers are an important source of information.
11. I intend to communicate and collaborate with other nodes in the blockchain network.
12. I often use keywords to help users locate shared information.
13. I believe that the information provided by the blockchain network is open and useful to the peers.

14. I'm interested in learning more about applications that can be developed in the blockchain networking environment.
15. I believe that blockchain technology will affect the way information is transferred between its peers.

The main observation could be that all blockchain users need to seek expansion and interoperability, and have the intention to share, discuss exchange information and gain knowledge. Furthermore, users intend to consider blockchain as an open network useful to their peers, develop applications and improve them, using state-of-the-art knowledge.

Accordingly, the second factor was described as "**Entrepreneurial and future prospects**" under the following items:

1. I believe that blockchain technology will affect tourism.
2. I believe that blockchain technology will affect the way a destination is advertised and promoted.
3. I believe that as technology evolves, more blockchain applications will be implemented in the global economy.
4. Blockchain technology will provide an opportunity for new start-ups to develop.
5. I am aware that blockchain technology has significant prospects in the near future.
6. I believe that the changes brought by blockchain technology will affect the way entrepreneurship evolves.
7. I believe that blockchain will provide entrepreneurial opportunities to future start-ups.
8. I believe that blockchain technology will affect the way of transactions are taking place.
9. The interoperability of blockchains will help boost entrepreneurship globally.

Respectively, blockchain users seem to have a subtle concern about future prospects and entrepreneurship. In fact, users believe that blockchain technology will affect several factors of the global economy and transactions as well, and will provide opportunities for future startups to develop and boost entrepreneurship globally.

Finally, the third factor extracted was "**Trust and security**", including the following items:

1. The blockchain network provides secure transactions.
2. I feel safe using the blockchain network.
3. I feel that other users also trust the blockchain network.
4. In a blockchain network I feel that my private data is not violated (I have confidence in the cryptography mechanisms).
5. I am aware of the Blockchain technology.

Regarding the third factor, blockchain users have a sharp interest in issues of trust and security. Likewise, they feel secure during transactions and knowledge exchange, and they also feel confident in cryptographic mechanisms.

The three factors generated by the confirmatory factor analysis describe the concern of global users towards the issues of expansion, improvement, interoperability, future prospects, entrepreneurship, trust and security.

## 5. Conclusions and Future Research

Blockchain technology is undoubtedly an innovative technology, which will radically change the business models of many industries, including tourism. This paper outlines the development and validation of a questionnaire aimed at measuring the level of knowledge spillover willingness within the tourism blockchain network. Based on the key components of the blockchain technology, which are decentralization, immutability, trust and security, the authors generated three factors, in order to identify knowledge spillover through a blockchain activated in the tourism industry.

The first factor concerning trust and security, which exists in a blockchain network, presents that cryptographic and consensus mechanisms enable nodes to interact with each other safely and without fear of breach of their personal data, as well as to rely on the trust that these mechanisms provide. Cryptographic mechanisms ensure secure transactions

with the result that there is trust between nodes, thus, it could be argued that trust is an important factor in knowledge spillover in a blockchain ecosystem.

The second factor, concerning entrepreneurship and the future prospects of blockchain technology, reflects the great influence that blockchain has on the evolution of the global economy. The importance of blockchain technology in entrepreneurship enables new businesses to operate in various industries, as well as in tourism industries adopting blockchain technology as their primary tool. This development of entrepreneurship clearly reflects the impact that blockchain technology will have on the future of the global economy, as well as on the development of the tourism sector. Given the aforementioned, future prospects and entrepreneurship are also significant factors in sharing knowledge in the blockchain network.

The third factor (network expansion, improvement and interoperability) is also an important factor for diffused knowledge within the blockchain environment. This transfer of knowledge by the nodes enables network scalability, as well as the participation of more nodes within blockchain networks. The scalability of the blockchain significantly affects tourism by expanding the industry globally and providing solutions regarding the cost and decentralization of transactions.

The use of blockchain in tourism could generate new business models and new forms of tourism, as well as a new form of tourists who transact, share information and interact with each other through the use of cryptocurrencies and blockchain networks. The blockchain provides nodes with access to an open and distributed information system, allowing nodes to access knowledge without interference from intermediaries. Likewise, access to knowledge without alteration by intermediaries maximizes the level of knowledge spillover, and this clear dissemination of knowledge without any alteration enables tourism and other industries the prospect of new business opportunities in the global economy.

It should be stated, however, that the limitations of this research could lie in the amount of correspondents and their familiarity with the continuing evolution of blockchain technology. Additionally, as well as the limitation regarded in terms of sample quality, the fact that the correspondents' vast majority were Greek also needs to be considered.

Future research is proposed, in order to extend this study further by investigating new prospects of the blockchain-related developments and their potential impact not only in the field of knowledge spillover but also in the tourism industry. By implementing new findings into the existing model, researchers can contribute to the creation of a new framework that thoroughly describes, analyzes and investigates the impact of blockchain on knowledge diffusion.

**Author Contributions:** Conceptualization, S.A. and P.M.; methodology, S.A and P.M.; software, S.A.; validation, S.A. and P.M.; formal analysis, S.A.; investigation, S.A.; resources, P.M.; data curation, P.M.; writing—original draft preparation, S.A.; writing—review and editing, P.M.; visualization, S.A.; supervision, S.A.; project administration, S.A.; funding acquisition, S.A. All authors have read and agreed to the published version of the manuscript.

**Funding:** This research received no external funding.

**Institutional Review Board Statement:** Not applicable.

**Informed Consent Statement:** Not applicable.

**Data Availability Statement:** The data presented in this study are available on request from the corresponding author. The data are not publicly available due to personal details of test participants.

**Conflicts of Interest:** The authors declare no conflict of interest.

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
