# Peer review of "Knowledge Spillover through Blockchain Network in Tourism: Development and Validation of Tblock Questionnaire"

_knowledge, doi:10.3390/knowledge2020019_

Round 1
Reviewer 1 Report
The paper will be interesting not only to readers who are dealing with the issues of ICT in tourism.
The abstract is informative but maybe it will make sense to add to the statement that “three factors revealed from the factor analysis, identifying the knowledge spillover from a blockchain network in tourism industry” the resulting statement what are these three factors specifically.
The keywords are in line with the terms used in the research. The references are correct of which most are up to date, and those from previous decades are relevant. The English language and style not bad, from my point of view, everything is clear and understandable.
The methodology is appropriate, widely used. The sample and the date of the field research are correctly detailed. I am not sure but maybe it will be better to state also the study hypotheses.
As to paper structure, the Introduction section correctly puts the research topic in context and the wording is appropriate and meaningful. I would only recommend to clearly state in this section the main purpose of the study (actually, the purpose stated twice in Section 3, lines 141-142, and 159-161; one of these sentences should be removed from Section 3 but can easily be used in the Introduction section). I also miss to the introduction a final paragraph about the paper structure.
There is the Results and Reliability Section and the structure of the section is a bit strange, four lines 181-184 for a subsection is too short, and lines 208-258 are not about factor analysis though put in the subsection 4.2. Factor analysis. At the same time, discussion is missing. It will be better to have a separate Discussion section, lines 208-258 and lines 267-298 could be moved there. Another option is to have a Results and Discussion section divided into two parts, first subsection on results and reliability, and the second subsection on discussion. In the Conclusion and Future research section, instead of lines 267-298, the authors could more thoroughly present their statement on contributions to the literature, practical implications, limitations of the research, as well as promising avenues for future research.
Author Response
Dear Reviewer,
First of all, we would like to thank you for your kind comments and the recommendation for our paper to be published. We have taken into account all of your comments regarding the article and we have tried to update the paper according to these comments. Regarding the comment on the hypothesis statement, we would like to underline that the purpose of the paper is to validate the questionnaire. This is why it is difficult for us to state the hypothesis.
Please find attached the updated paper according to the comments.
Thank you in advance.
Kind regards,
Spyros Avdimiotis and Panagiotis Moschotoglou
Reviewer 2 Report
- The literature is not sufficiently referenced.
- The pattern from the conclusions should be moved to the results. The results should be strengthened.
- What is the state of research in the field in Greece?
- The objectives must be more clearly stated, as well as the limitations of the study.
- The appendix must be included in the text.
- More data about questionnaire are required.
Author Response
Dear Reviewer,
First of all, we would like to thank you for your kind comments and the recommendation for our paper to be published. We have taken into account all of your comments regarding the article and we have tried to update the paper according to these comments. Regarding the current research situation in Greece and especially in the field of the tourism industry, we would like to inform you that this paper and this questionnaire is the first research attempt regarding the research part of the implementation of blockchain technology and knowledge spillover.
Please find attached the updated paper according to the comments.
Thank you in advance.
Kind regards,
Spyros Avdimiotis and Panagiotis Moschotoglou
Round 2
Reviewer 1 Report
The reviewer's comments are taken into account.
Author Response
Dear Reviewer,
We would like to thank you for your recommendations and your kind comments. Your contribution during the review process was important and decisive.
Thank you once again.
Kind regards,
Dr. Spyros Avdimiotis and Mr. Panagiotis Moschotoglou